# Growth of *Aspergillus fumigatus* in Biofilms in Comparison to *Candida albicans*

**DOI:** 10.3390/jof8010048

**Published:** 2022-01-04

**Authors:** Eefje Subroto, Jacq van Neer, Ivan Valdes, Hans de Cock

**Affiliations:** Molecular Microbiology Laboratory, Department of Biology, Faculty of Science, Utrecht University, Padualaan 8, 3584 CH Utrecht, The Netherlands; ekdsubroto@gmail.com (E.S.); j.f.vanneer@uu.nl (J.v.N.); I.D.ValdesBarrera@uu.nl (I.V.)

**Keywords:** *Aspergillus fumigatus*, biofilm, in-host adaptation, gene expression

## Abstract

Biofilm formation during infections with the opportunistic pathogen *Aspergillus fumigatus* can be very problematic in clinical settings, since it provides the fungal cells with a protective environment. Resistance against drug treatments, immune recognition as well as adaptation to the host environment allows fungal survival in the host. The exact molecular mechanisms behind most processes in the formation of biofilms are unclear. In general, the formation of biofilms can be categorized roughly in a few stages; adhesion, conidial germination and development of hyphae, biofilm maturation and cell dispersion. Fungi in biofilms can adapt to the in-host environment. These adaptations can occur on a level of phenotypic plasticity via gene regulation. However, also more substantial genetic changes of the genome can result in increased resistance and adaptation in the host, enhancing the survival chances of fungi in biofilms. Most research has focused on the development of biofilms. However, to tackle developing microbial resistance and adaptation in biofilms, more insight in mechanisms behind genetic adaptations is required to predict which defense mechanisms can be expected. This can be helpful in the development of novel and more targeted antifungal treatments to combat fungal infections.

## 1. Introduction

*Aspergillus fumigatus* (*A. fumigatus*) is a saprophytic fungus that can normally be found in the soil, where it lives on organic debris. The airborne conidia of this opportunistic fungus are spread abundantly, and hundreds of conidia are inhaled by humans on a daily basis [1]. Inhalation of these conidia by healthy individuals is normally not harmful since they are eliminated by lung innate immune defense systems like mucociliary clearance [2] and alveolar macrophages, the main phagocytic cells of the respiratory tract [3]. However, *A. fumigatus* can cause infections in the respiratory tract of immunocompromised individuals, such as patients following chemotherapy, receiving transplants and immune-suppressive drugs or those in intensive care [4,5]. In most of these patients, *A. fumigatus* infection starts in the respiratory tract, resulting in varying clinical diseases such as invasive aspergillosis (IA), allergic bronchopulmonary aspergillosis (ABPA) and aspergilloma [1]. Next to *Candida albicans* (*C. albicans*), *A. fumigatus* is also a major pathogen responsible for nosocomial infections. Whereas *Candida* species are responsible for 80% of fungal nosocomial infections [6], *A. fumigatus* is the most common species to occur in infections in the airways of immunocompromised individuals, causing severe and fatal invasive infections [1]. 

Developments in medical therapy and technology such as organ transplantations accompanied with immunosuppression and new chemotherapeutic agents have increased the survival of patients suffering from severe diseases that previously would be fatal. Due to these advances in medical technology, the number of immunocompromised patients that are vulnerable to nosocomial infections by fungi has also increased [7]. In hospitals, nosocomial aspergillosis has become a severe threat to immunosuppressed patients due to the airborne conidia of *A. fumigatus*, and where outbreaks of nosocomial aspergillosis can occur due to construction work or deficient ventilation systems [8,9]. Moreover, the implementation of medical devices such as prostheses, catheters, and mechanical heart devices are also a major source of fungal as well as bacterial infections that can be difficult to treat [10,11]. 

Infection of *A. fumigatus* can be severe and difficult to treat due to its ability to form a biofilm on surfaces [12]. These biofilms consist of a microbial community that can strongly adhere to each other and to biotic and abiotic surfaces. The microbial community is encased by a polymeric extracellular matrix (ECM), that is composed of primarily of polysaccharides. The ECM functions as a protective sheet and offers a framework for cell cohesion and adhesion to the surface [13]. Moreover, the ECM controls dispersion of cells and acts as a source of nutrients for the cells. The shield of biofilm also possesses defensive characteristics, making the biofilm more tolerant against immune cells and drug treatments [10]. 

To develop strategies to prevent or treat infections of *A. fumigatus*, the underlying mechanism of biofilm formation needs to be understood. This paper will give an overview of the current knowledge on *A. fumigatus* biofilm formation compared to the knowledge available for *C. albicans* biofilm, which is studied more broadly. First, the different stages and mechanisms of biofilm formation will be described and compared with the known mechanisms of *C. albicans*. Secondly, the interaction between *A. fumigatus* and other microorganisms will be highlighted. Finally, the mechanism and effect of genetic adaptation and micro-evolution in *A. fumigatus* biofilms will be described. 

## 2. Stages of Biofilm Development and Molecular Pathways

The formation of a fungal biofilm starts with the attachment of conidia to the biotic or abiotic surface. In vitro studies using scanning electron microscopy (SEM) and confocal laser scanning microscopy [14,15] showed that the formation of *A. fumigatus* biofilm can be roughly divided into different stages; (1) adhesion to the surface, (2) conidial germination into hyphae and development, (3) biofilm maturation with development of mycelia, ECM production, hyphal layering networks and formation of channels, and (4) cell dispersion [15]. In *C. albicans*, the stages of biofilm development are similar to *A. fumigatus* and comprise adhesion, initiation of biofilm formation, biofilm maturation and dispersion [16].

### 2.1. Adhesion

Adhesion is considered to be an important step in the process of fungal infection, since it is the first interaction between the fungi and the host. After inhalation, the airborne conidia come in contact with the respiratory tract and adhere onto airway epithelial cells [17]. The initial attraction between the conidia and surface is weak, and to strengthen attachment to the host surface, cell-surface components known as adhesins are used in fungal biofilm formation [15]. In the biofilm formation of *C. albicans*, fungal adhesion is regulated by members of a class of proteins known as glycosylphosphatidylinositol-dependent cell wall proteins (GPI-CWP). These GPI-CWP proteins include Hwp1, Hwp2 and a group of eight adhesion genes (*ALS*1-*ALS*7 and *ALS*9) that are part of the agglutinin-like sequence (*ALS*) family. These genes encode for proteins that possess characteristics of adhesin glycoproteins on the cell surface [18,19,20]. 

In *A. fumigatus*, electron microscopic and biochemical studies have identified molecules on the outer layer of the cell wall of *A. fumigatus* conidia. This study showed that the outer layer of the conidia was characterized by a layer of rodlets made of hydrophobins, a family of small hydrophobic proteins on the cell surface [21]. These hydrophobins were identified as RodAp, RodBp, RodCp, RodDp, RodEp and RodFp [15]. It is suggested that the hydrophobic characteristic of these proteins allows for their adhesion to hydrophobic abiotic or biotic surfaces, and that they are involved in the initiation of the biofilm process [22]. In a study where single and multiple hydrophobin-deletion mutants were constructed, the results showed that only RodA is necessary for efficient rodlet formation on the cell surface of conidia and adherence on polystyrene plates [23]. RodA proved to be responsible for hydrophobicity, integrity of the cell wall, conidial production, and susceptibility to external factors such as drugs. The importance of RodA was also observed in another study, where disruption of RodA decreased adherence of conidia to collagen and albumin, although not laminin or pulmonary epithelial cells [17]. The rodlet layer that includes RodA, also disguises the fungal cell wall components β-glucan, chitin, and glycoproteins that can provoke immunogenic reactions. Whereas β-glucan is recognized by receptors such as Dectin-1 in case of interaction of epithelial cells, interaction with ephrin type-A receptor 2 is also involved [24,25]. Another component that is suggested to be involved in the adhesion process of *A. fumigatus* is sialic acid. This component is part of a family of derivatives of neuraminic acid present on the cell surface of conidia. Removal of the surface sialic acids on the conidial surface resulted in decreased adherence of the conidia to fibronectin, suggesting that conidia adhere to basal lamina proteins via the negatively charged sugars on their surface, which are presumed to be sialic acids [26]. 

Other cell wall carbohydrate components also showed to mediate fungal adherence. Purified fragments of fibronectin were used to analyze which components on the conidial surface are involved in the process of adherence. The results showed that *A. fumigatus* conidia preferably bound to the non-glycosylated 40-kDa fragment, containing the glycosaminoglycan binding domain. Negatively charged carbohydrates, such as dextran sulfate and heparin, as well as high-ionic strength buffers, hindered binding of conidia to both fibronectin and basal lamina. This indicated that negatively charged carbohydrates on the cell wall surface of conidia may bind to the glycosaminoglycan binding domain of fibronectin and other basal-lamina proteins. Together, these data suggest that negatively charged carbohydrate moieties on the conidial surface could mediate binding to macromolecules of the host [27].

The developmentally regulating protein MedA was proposed to play an important role as an adhesin and in regulating the adherence and expression of conidiation genes. *A. fumigatus* mutants deficient in MedA were impaired in adherence to pulmonary epithelial cells, vascular endothelial cells and fibronectin [28]. Next to fungal adhesion to host cells, MedA also manages efficient biofilm formation, since MedA deletion mutants also showed to be decreased in biofilm formation. However, the exact molecular mechanism whereby MedA enhances adherence remains unclear. 

At a later stage during biofilm formation, adhesion can also occur on a deeper level in the human host. Growing hyphae can interact with pulmonary epithelial cells, where they adhere to and invade the abluminal surface of vascular endothelial cells to eventually access the vascular compartment. When the hyphae entered the blood vessels, hyphal fragments can be distributed to other sites in the host where they can adhere to the luminal surface of endothelial cells before passing through them and invading into deep tissues. *A. fumigatus* hyphae can induce host cell damage and death. There, it is likely that the basal lamina within the airways and blood vessels are exposed, and that fungal cells can adhere to basement membrane macromolecules including laminin, fibronectin and collagen [17]. 

### 2.2. Conidial Germination and Hyphal Development

After attachment of *A. fumigatus* conidia to the surface, further colonization occurs through hyphal proliferation [14]. Additionally, as shown in a study by Wasylnka & Moore [29] with two *A. fumigatus* strains expressing green fluorescent protein, conidia can be internalized by nonprofessional phagocytes in vitro such as epithelial and endothelial cells to germinate at a later stage [30]. This was also observed with conidia of *C. albicans*, where a study showed that blastoconidia could be internalized by endothelial cells and macrophages in mice [31]. 

The conidial germination of *A. fumigatus* involves disruption of the hydrophobic rodlet layer, revealing the inner conidium walls that are composed of polysaccharides, which are hydrophilic cell wall components. During germination, a hydrophobic tip can be found on a single germinating spore. The conidium loses its surface hydrophobicity progressively and the new growth-point exhibits a coexistence of hydrophobic rodlets and hydrophilic polysaccharides [15,22]. 

After adhesion of the conidia, conidial germination into hyphae begins with the formation of tube-like channels that possess the hydrophobic nature of the cell wall that is suggested to enhance hyphal development [22]. In hyphal development, the adhesive exopolysaccharide galactosaminogalactan (GAG) plays an important role in adherence of *A. fumigatus* hyphae to cells of the human host and modulating the immune response during infection. GAG is secreted by growing hyphae, where it binds to the surfaces of these hyphae, resulting in a polysaccharide sheath covering the growing organisms [32]. The fungal regulatory protein MedA and developmental transcription factor StuA both affect the formation of adherent biofilms. Disruption of StuA in *A. fumigatus* strains resulted in reduced adherence to pulmonary epithelial lines and other abiotic substrates [33]. A comparative transcriptome analysis of the Δ*medA* and Δ*stuA* regulatory mutants identified a gene encoding a putative UDP-glucose-epimerase, designated *uge3*, which was dysregulated in both the Δ*stuA* and Δ*medA* mutants. Disruption of Uge3 resulted in full impairment of GAG synthesis, and markedly decreased adhesion to host cells and biofilm formation. This indicates that GAG is necessary for the efficient attachment of hyphae and biofilm structural integrity [33].

In *C. albicans*, a proliferation phase is also followed after attachment of the yeast cells, and initiation of filamentation will result in hyphal development, eventually leading to the biofilm development process [34]. In hyphal development, hyphal morphogenesis is important for adhesion maintenance. In this process, the proteins EFG1 and BCR1 are key regulators of several adhesin genes, including most of the ALS gene family, EAP1 and HWP1. A study by McCall et al. [35] showed that adhesion maintenance proteins in *C. albicans* are expressed at different times during growth phases. Whereas the adhesion maintenance protein Ywp1 is expressed during late biofilm growth, disruption of EFG1 and BCR1 in *C. albicans* deletion strains demonstrated poor adherence of hyphae in biofilm. This indicates that filamentation is important for further expression of adhesion maintenance proteins. Moreover, the expression of *BCR1* is under the regulation of the hyphal regulator Tec1p, where *BCR1* is a downstream component of the hyphal regulatory network that couples expression of cell-surface genes to hyphal differentiation. This indicates that hyphal cells are specialized to present adherence components to ensure biofilm structure and integrity [36]. 

### 2.3. Biofilm Maturation

After development of the hyphae, the process of biofilm maturation is initiated. During this stage, a biofilm matrix is formed consisting of the extracellular matrix (ECM) that covers the colony surface and binds the cells to form the structural base of the biofilm and glues the hyphae together [12,15]. In *A. fumigatus*, the structural components of the mature biofilm generally consist of GAG, galactomannan, α-1,3 glucans, monosaccharides, proteins, antigens and lipids, melanin, polyols, and extracellular DNA (eDNA) [13]. Additionally, an expression study showed that there are at least two hydrophobin genes (*rodB* and *rodC*) expressed in aerial static mycelium. However, no rodlet proteins were detected in the same ECM [12]. 

In *C. albicans*, the biofilm matrix is characterized by a structured mixture of pseudohyphal and hyphal cells encased by an ECM that consists of glycoproteins, carbohydrates, α-mannan and β-1,6-glucan polysaccharides, β-1,3-glucan, lipids and eDNA [16]. 

The structure of biofilms is dependent of the environmental conditions of the microorganism. Therefore, the biofilm structure can vary and keep reshuffling as they keep on adapting to their environment. For example, the composition of the ECM could vary between aspergillosis pathologies. In a study by Loussert et al. [37] it was shown that an ECM is produced in the lungs at the surface of *A. fumigatus* hyphae that are present in patients suffering from aspergilloma or IA. The ECM at these different sites showed a different hyphal organization. Light microscopic observations showed that during aspergilloma a ball of strongly agglutinated hyphae without any host cells inside was formed. Additionally, a dense material surrounded the hyphae, and at the periphery of the ball, many blood cells were identified. On the contrary, the hyphae detected in experimental IA were shown to be separated. Moreover, the ECM at these sites contained galactomannan and GAG as observed in vitro. Nevertheless, α-1,3 glucan which was also present in the ECM in vitro was detected only in aspergilloma at the periphery of the ECM. Other components, such as the major antigenic glycoproteins, were present in aspergillosis and IA ECM in vitro but were not observed in the ECM produced in vivo. A possible explanation for this effect is that these antigens are secreted during infection, but do not accumulate as much as seen in vitro, since these antigens were detected in patients with aspergillosis and IA [37].

In fungal biofilm development, the ECM that is formed contributes to the structural base of the biofilm, including the exopolymeric substance (EPS) and network of mycelia. The EPS occurs with a mucous appearance that adheres completely and covers the hyphae, forming a covering sheath over the fungal cells [15]. In the mature biofilm, the immobilized cells function as an ecosystem that includes continuous interactions in the form of cell-cell communication acting as a structured network that holds the cells together. The EPS in the ECM undertakes many important functions such as adhesion, cell aggregation, providing protection against antimicrobial agents and host defense mechanisms, providing a nutrient source, stabilization of the cell community and the exchanging of genetic information [15]. 

In both *A. fumigatus* and *C. albicans*, eDNA is an important component of the ECM, where it contributes to structural integrity and antifungal resistance [38,39,40]. A mechanism by which eDNA can provide biofilm resistance is by changing the extracellular environment. Since eDNA is an anionic macromolecule, this molecule is able to chelate cations such as magnesium ions, resulting in a decline of effective concentration of Mg^2+^ in the fungal environment. Moreover, certain concentrations of DNA can regulate the induction of the cationic antimicrobial peptide resistance operon, *PA3552–PA3559* in *Pseudomonas aeruginosa* (*P. aeruginosa*). This indicates that the presence of eDNA can contribute to genomic DNA release and inducible antibiotic resistance [41]. 

Another role of eDNA is identified as providing the exchange of genetic information by horizontal gene transfer [42]. Several studies showed that plasmid transfer occurs at an increased rate in biofilms compared to planktonic cultures. A study by Hausner & Wuertz [43] quantified the gene transfer process in a simple laboratory-based biofilm system with a strain of *Alcaligenes eutrophus* as a recipient of a green fluorescent protein-tagged plasmid derivative from *Escherichia coli* (*E. coli*). The results of this experiment showed that more transconjugants occurred compared to on a plate count. The transfer of genes in eDNA can contribute to the development of antibiotic resistance in microbial populations. 

In *A. fumigatus*, the eDNA is created by autolysis and derives from fungal cells by the secretion of chitinases by *A. fumigatus*, favoring its release [39]. Although it is not yet clear how eDNA contributes to antifungal resistance, treatment of *A. fumigatus* biofilm with DNase resulted in a destabilized integrity of the biofilm and an increased susceptibility to antifungals amphotericin B and caspofungin. Therefore, it is suggested that the presence of eDNA reduces exposure of the target cells to the antifungal drugs [13]. 

In *C. albicans* as well as *A. fumigatus*, a large part of the ECM consists of components of the host when grown in vivo. A previous study showed that in three infection site models of *C. albicans,* a biofilm matrix was produced containing host components such as hemoglobin, albumin, and alpha globulins, amylase, fibrinogen and keratin. The presence of host components also depended on the biofilm surface (venous catheter, urinary catheter or denture model). Although the precise role of these host components in the biofilms are not yet clear, the attraction of blood cells could be beneficial in the acquisition of iron. Moreover, another explanation could be that the host components are used for the structure of the biofilm to save energy by producing these components themselves [44,45]. 

### 2.4. Cell Dispersion

The stage of cell dispersion is generally considered as the terminal stage of biofilm development, also referred to as seeding dispersal [46]. Cell dispersion typically occurs in response to stressful environmental changes, where dissemination allows viable cells to persist in other locations of the host where it can further reproduce [15]. In cell dispersion of *A. fumigatus,* the part comprising the conidia or hyphae are detached from the biofilm. In the study of González-Ramírez et al. [15], asynchronous biofilm development was observed, in particular at the biofilm-maturation stage when the new conidia were capable of germinating, producing new mycelial growth and hyphal modifications prior to dispersion. In *C. albicans*, the cells dispersed from a biofilm are yeast cells that originate from the topmost hyphal layers of the biofilm [47]. In *C. albicans*, the dispersion of round, yeast-form cells are suggested to occur during biofilm development, but larger amounts of yeast cells are dispersed when the biofilm is in a more mature stage. Cell dispersion can lead to novel biofilm formation in the host, possibly leading to systemic infections in the bloodstream or to the spreading of the infection to other parts of the host. Known transcriptional regulators that are involved in *C. albicans* cell dispersion are Ume6, Nrg1 and Pes1 (also known as Nop7). Also, a component of a chromatin-modifying complex, known as Set3, is required for dispersal. Set3 is suggested to be recruited by the transcriptional regulator Nrg1 [16]. The heat shock protein 90 (Hsp90) is suggested to play a role in cell dispersion of *C. albicans*. Experiments performed in vitro showed that inactivation of Hsp90 reduced *C. albicans* biofilm growth and maturation and reduced dispersal of biofilm cells [48]. Additionally, Hsp90 plays a role in the stress response pathways, which is discussed in the next section. If Hsp90 also plays a role in the process of *A. fumigatus* cell dispersion is not known.

## 3. Drug Resistance Mechanisms

One of the most important characteristics of the biofilm is to function as a protective shield, making the community more tolerant against physical disruptions and drug treatments. *A. fumigatus* and *C. albicans* have several resistance mechanisms to protect them against environmental pressures and the in-host environment. An overview of these mechanisms is given in Table 1. In both *A. fumigatus* and *C. albicans*, it is hypothesized that the presence of the ECM reduces drug susceptibility by working as a shield that absorbs the drug and inhibits it from reaching its cellular targets. In a study by Al-Fattani and Douglas [49], a *C. albicans* biofilm was examined under continuous flow resembling in vivo conditions, and a correlation between the degree of extracellular matrix production and antifungal resistance was found. For *A. fumigatus*, the formation of biofilm is also suggested to hinder or delay penetration of antifungal agents, but the underlying mechanisms remains unclear [12,40]. Other mechanisms in *A. fumigatus* and *C. albicans* to promote drug resistance are the use of multidrug resistance (MDR) pumps, also known as efflux pumps, which transport toxic substrates from the cells into their environment. The expression of MDR pumps in *A. fumigatus* (AfuMDR) was examined in a study by Rajendran et al. [50], where in an in vivo mouse biofilm model, constitutive upregulation of *AfuMDR4* was detected after treatment with voriconazole after 24 h. This suggested that efflux pumps were expressed in *A. fumigatus* biofilms, and thus contributes to azole resistance. Other known efflux pumps contributing to the transport of toxic components are MDR1, MDR2, and MDR4 [51]. Quorum sensing mechanisms could affect biofilm formation and development of drug resistance in *A. fumigatus* biofilms, although this has hardly been investigated. In this respect the inhibitory effects of farnesol on *A. fumigatus* growth by misplacing the tip localized Rho1 and Rho3 proteins and inhibition of cell wall signaling are remarkable [52]. Farnesol was previously shown to reduce the transition to hyphal forms and activated the expression of drug-resistance genes of *C. albicans* [53]. Furthermore, it was shown that farnesol affected biofilm formation of *C. albicans* and was a specific modulator of drug efflux pumps of the ABC-multidrug family [54]. 

For *C. albicans*, a study by Mukherjee et al. [55], showed that efflux pumps CDR1, CDR2 and MDR1 contribute to azole resistance. However, this effect was only shown in the early-phase biofilms but not the mature biofilm phase, since knock-out mutants showed to be more susceptible to the azole in early-phase biofilms. Disruption of these efflux pumps did not affect biofilm formation. Furthermore, variation in the sterol profile as a possible mechanism of drug resistance was investigated. Sterol analysis showed that changes in the composition of sterol are involved in the intermediate and mature phases of biofilm development in *C. albicans*.

Next to contributing to biofilm growth, integrity and maintenance, eDNA was found to play a role in promoting drug resistance. As mentioned earlier, degradation of eDNA by DNase resulted in loss of biofilm mass and increased susceptibility to antifungals in *A. fumigatus* and *C. albicans* [38,39]. Moreover, eDNA induces conidial surface adhesion in early stages of biofilm development and co-localizes with the polysaccharides of the ECM biofilm, and becomes part of the ECM surrounding the biofilm cells. The eDNA is suggested to enhance the structure of biofilms, but the exact mechanisms behind this effect remains unclear. Therefore, it would be interesting to analyze whether eDNA that originates from neutrophils serves as a crucial nutrient source for the fungal biofilm and is beneficial to biofilm formation in vivo [45]. 

A resistance characteristic of *A. fumigatus* and *C. albicans* biofilm against antimicrobial agents are a special class of protected cells also known as persister cells. These persister cells are hypothesized to be cells that have differentiated into an inactive but protected state and neither grow nor die in the presence of microbicidal antibiotics. Persister cells are dormant variants of regular cells and are randomly formed in biofilms. These cells are found in *A. fumigatus* and *C. albicans* biofilms, and hold high tolerance against antifungals [56]. In the study of Lafleur et al. [57], treatment of *C. albicans* biofilms with two antimicrobial agents, amphotericin B and chlorhexidine, resulted in a biphasic killing pattern, indicating that a subpopulation of highly tolerant cells, termed persisters, existed. The surviving *C. albicans* persisters were only detected in biofilms, in contrast to planktonic populations. The persister cell hypothesis could explain why biofilms are protected from a wide set of antimicrobial agents. Moreover, reinoculation of the cells that survived the killing of the biofilm by amphotericin B produced a new biofilm with a new subpopulation of persisters. This indicated that *C. albicans* persister cells are not mutants but phenotypic variants of the wild type [57]. The population of persister cells has been estimated to form about 0.1–10% of all cells in the biofilm and are suggested to survive exposure to antimicrobial agents and reseed the biofilm. Persister cells have been identified in several pathogens such as *E. coli*, *Staphyloccoccus aureus* (*S. aureus*) and *P. aeruginosa*, where large populations of persisters were identified [58]. 

Certain stress pathways can be induced in the biofilm to enhance antimicrobial resistance. In *C. albicans*, the component calcineurin, a Ca^2+^ calmodulin-activated serine/threonine-specific protein phosphatase, plays an important role in homeostasis, morphogenesis, virulence, and was also found to play a role in resistance development. Disruption of this component was shown to improve the efficacy of fluconazole on *C. albicans* biofilm [59]. Next to its role in cell dispersion, the heat shock protein Hsp90 is also involved in another *C. albicans* stress response pathway. Hsp90 has been shown to enable the emergence and maintenance of drug resistance in planktonic conditions by stabilizing the protein phosphatase calcineurin and MAPK Mkc1. Next to reduced biofilm growth, maturation and cell dispersion, inactivation of Hsp90 also impairs resistance of *C. albicans* biofilm to antifungal drugs such as azoles [48]. Also in *A. fumigatus*, inhibition of Hsp90 in vitro was shown to improve the efficacy of antifungals such as azoles and echinocandins against biofilms. This implicated that Hsp90 acts as a regulator in drug resistance [48]. 

An additional component involved in the stress response pathway includes the MAPK MKc1p, an important component of the signal transduction pathway that is triggered by cell wall stress. Inactivation of this protein resulted in defective hyphal formation and the development of biofilm and resistance to fluconazole in *C. albicans* mutants [60].
jof-08-00048-t001_Table 1Table 1Drug resistance mechanisms in *A. fumigatus* and *C. albicans* biofilm.
*A. fumigatus**C. albicans*ReferencesECMReducing drug susceptibility by preventing the drug from reaching their cellular targetReducing drug susceptibility by preventing the drug from reaching their cellular target[12,49]Efflux pumpsUpregulation of efflux pumps genes such as AfuMDR4, MDR1, MDR2, MDR4CDR1, CDR2 and MDR1 contributing to azole resistance[40,51]eDNAPromotes resistance against amphotericin B and caspofunginsPromoting resistance to amphotericin B and echinocandins[39,40]Persister cellsDormant drug tolerant cells developed during biofilm formation that can serve as an inoculum for new biofilmsDormant drug tolerant cells developed during biofilm formation that can serve as an inoculum for new biofilms[40,56]Induced stress response pathwayHSP90 pathwayMAPK-, HSP90- and calcineurin pathway to develop azole resistance [40,48]

## 4. Interaction with Host Immune Systems

Next to drug resistance mechanisms, another essential characteristic of the biofilm is the ability to evade or affect the immune systems of the host. Whereas *A. fumigatus* and *C. albicans* share similar mechanisms in drug resistance, the mechanisms to evade host immune systems are more different. 

In *A. fumigatus*, the exopolysaccharide GAG is secreted by the growing hyphae and plays several roles in the interaction between the host and pathogen (Figure 1). It functions as an adhesin and interferes with immune recognition of the host by reducing exposure of the cell wall components like β-1,3 glucans that are considered important fungal pathogen-associated molecular patterns (PAMPs) recognized by the innate immune system [32]. The β-1,3 glucans are increasingly exposed as conidia swell and begin to germinate, but are covered again during the production of GAG during hyphal growth. Although it is likely that there are more PAMPS that are covered by GAG, these are not identified yet. GAG also interferes with neutrophil-mediated immunity to *A. fumigatus*, where it induces neutrophil apoptosis by triggering neutrophils to produce reactive oxygen species (ROS). Moreover, GAG also impedes neutrophils through extracellular killing by neutrophil extracellular traps (NETs). Additionally, GAG is able to induce immunomodulatory effects by modulating the cytokine production and T-cell responses [32]. 

The extracellular matrix of *C. albicans* biofilms are crucial in the protection against the host immune cells. For example, ROS and NETs are less triggered by *C. albicans* biofilms compared to planktonic cells [61,62]. Time lapse imaging showed impaired NET release during growth of *C. albicans* biofilms. This revealed that NET inhibition depended on an intact ECM since genetic or physical disruption of this component resulted in NET release. This showed that *C. albicans* biofilms can hinder neutrophil response through an inhibitory pathway induced by the ECM [61]. Additionally, the polysaccharide β-1,3 glucan in the biofilm matrix is suggested to prevent drugs from reaching their targets, but also to reduce neutrophil activation. In a study by Xie et al. [63] it was shown that the presence of extracellular polysaccharides such as β-glucans can inhibit neutrophil antibiofilm function. Early biofilms were coated with glucanase-treated matrix or laminarin, which showed that some β-glucans interfere with neutrophil–biofilm interactions that activate an ROS response. Although components on the surface of the biofilm matrix can interfere with the pattern recognition receptors in the host immune cells, the exact mechanisms underlying the interaction between biofilms and the host immune systems still need to be investigated further.

## 5. Interaction between *A. fumigatus* and Other Micro-Organisms

In nature, biofilms are generally composed of several micro-organisms. Co-infections of interacting microorganisms of varying species can form an ecosystem of pathogens leading to worsened disease in the patient. For example, co-infection of *C. albicans* with other pathogens such as *E. coli* can enhance the adhesion of *C. albicans* in bladder mucosa. Similarly, pneumonia caused by *P. aeruginosa* infection is more likely to occur in individuals already infected by *C. albicans* [64]. Co-existence of bacteria and fungi in the human host can result in co-aggregation and form mixed-species biofilm. This can lead to an even more harmful infection in the human host than infection of a single pathogen. 

Whereas interaction of *C. albicans* with other microorganism species are more widely studied [16], interaction of *A. fumigatus* with other microbial species is less described, except for interaction with *P. aeruginosa* and *S. aureus*. Interaction between *A. fumigatus* and *S. aureus* is observed in the eye disease keratitis. However, this interaction is scarcely studied, but analysis of the mixed biofilm showed structural changes of both microbes and an antibiosis effect of *S. aureus* on *A. fumigatus* was observed and which strongly affected hyphal development [65]. More information is available regarding the interaction between *A. fumigatus* and *P. aeruginosa* and which will be described in the following sections. In patients suffering from cystic fibrosis (CF), *A. fumigatus* and *P. aeruginosa* are the species that are found most commonly [66,67]. Co-colonization of these opportunistic pathogens can lead to both mutual inhibition and promotion of their growth in biofilm. 

### 5.1. Inhibitory Mechanisms of P. aeruginosa on A. Fumigatus

In vitro, growth of *A. fumigatus* can be inhibited by *P. aeruginosa* by the effect of phenazines, a nitrogen-containing heterocyclic compound secreted by *P. aeruginosa* during growth. This compound is able to affect the modification of multiple host cellular response mechanisms, and is considered an important antimicrobial factor against bacteria, fungi or mammalian cells, and is involved in the interactions with eukaryotic hosts and host tissues [68]. The *P. aeruginosa* phenazines are involved in electron shuttling, redox chemistry, and the promotion of *P. aeruginosa* biofilm growth through toxic superoxide generation and signaling. Phenazines include five secreted molecules; 5-N-methyl-1-hydroxyphenazine (PYO), 1-hydroxyphenazine (1-HP), phenazine-1-carboxamide (PCN), phenazine-1-carboxylic acid (PCA) and dirhamnolipids (diRhls) [69]. The secreted phenazines inhibit *A. fumigatus* growth by generating reactive oxygen species (ROS) and reactive nitrogen species (RNS). Whereas PYO, 1-HP, PCN and PCA inhibit *A. fumigatus* growth through RNS/ROS, DiRhls can inhibit *A. fumigatus* biofilm growth by blocking the β-1,3 glucan synthase activity in *A. fumigatus* [70]. An overview of different interaction mechanisms between these two pathogens is described by Zhao & Yu, [69]. An important note is that the studies that show the inhibitory effect of phenazines on *A. fumigatus* were performed in vitro. To analyze if the same effects are also present in vivo, further research is needed [69]. 

One of the key mechanisms of *A. fumigatus* inhibition by *P. aeruginosa* involves the competition for iron. To survive, both microbes are dependent on iron in their environment. *P. aeruginosa* produces the siderophore pyoverdine that is able to chelate iron and deny this molecule to *A. fumigatus*. In turn, *A. fumigatus* uses hydroxamate siderophores (*sidA*) to defend itself against these antifungal effects of *P. aeruginosa* [71]. Another study showed that the *Pseudomonas* quinolone signal (PQS), a quorum sensing molecule, is also able to chelate iron and delivers iron to the *P. aeruginosa* cell membrane using siderophores [72]. The bacteriophage Pf4 produced by *P. aeruginosa* also binds iron and thereby causes an iron deficiency for *A. fumigatus.* However, inhibition by Pf4 was shown to be more effective in the developing stage of *A. fumigatus* biofilm than in biofilm formation [73]. Small colony variants (SCVs) of *P. aeruginosa* are slow growing sub-populations of bacteria with different phenotypic traits such as an atypical colony morphology and different biochemical characteristics. Therefore, they are more difficult to identify clinically. These SCVs are more adapted to persist in mammalian cells and are less vulnerable to antimicrobial agents than the wild type colonies. They can cause recurrent or latent infections in case of an emerging protective environment of the host cell [74]. The SCVs formed by *P. aeruginosa* are involved in the inhibition of *A. fumigatus* biofilm formation. This inhibition is related to the pyoverdine pathway [75].

### 5.2. Inhibiting Mechanisms of A. fumigatus on P. aeruginosa

*A. fumigatus* has developed mechanisms to withstand the growth-inhibitory effects of *P. aeruginosa.* For example, the secreted bacterial phenazines can be modified by *A. fumigatus* metabolically and decrease their inhibitory effect. The phenazine PCA can be modified to 1-HP,1-methoxyphenazine and phenazine-1-sulfate. As described previously, 1-HP has an inhibitory effect on *A. fumigatus*. However, 1-HP is also able to induce the production of the *A. fumigatus* siderophores TAFC and FsC. The phenazines PCA and PYO produced by *P. aeruginosa* can be converted by *A. fumigatus* to phenazine dimers, which have a decreased inhibitory effect on *A. fumigatus* biofilm growth. PCA also plays a role in the iron acquisition of *P. aeruginosa* and conversion of PCA can lead to reduced biofilm formation. PYO is a QS molecule and affects transcriptional regulation and induces biofilm formation. Therefore, conversion of PYO might negatively affect the QS regulation system [69]. 

Gene expression and proteomic analyses showed that in a mature biofilm, specialized genes for the production of gliotoxin are expressed [76]. Gliotoxin secreted by *A. fumigatus* is an immuno-evasive toxin that plays an important role in mediating *A. fumigatus*-associated colonization. Production of gliotoxin is upregulated during mycelial growth in *A. fumigatus* and in biofilms. The increased biofilm production of *A. fumigatus* also helps to inhibit *P. aeruginosa* growth. The hyphae present in the *A. fumigatus* biofilm are more mature with a different morphology, and it is suggested that these mature hyphae are less susceptible to the cytotoxic molecules produced by *P. aeruginosa* compared to germinating conidia and germlings. The increased sensitivity of the latter for *P. aeruginosa* virulence factors was proposed to be due to a difference in the metabolic activity which is uniform in germlings but more localized to apical regions in mature hyphae [77]. 

### 5.3. Mutualism between P. aeruginosa and A. fumigatus

*P. aeruginosa* can promote the growth of *A. fumigatus* under certain conditions. Although secreted phenazines of *P. aeruginosa* inhibit growth of *A. fumigatus*, sub-bacteriostatic concentrations of phenazines can promote growth in *A. fumigatus* under specific circumstances. For example, the sub-bacteriostatic phenazines PYO, PCA and PCN can promote *A. fumigatus* growth by enhancing iron uptake. The redox function of 1-HP which is able to chelate iron and deprive *A. fumigatus* of iron can also promote *A. fumigatus* growth [78]. 

Volatile organic compounds (VOCs) secreted by *P. aeruginosa* such as volatile sulfur compounds (VSCs) can promote *A. fumigatus* growth in vitro [79]. Via genetic approaches and gas chromatography mass spectrometry-mediated volatile analysis, it was shown that *A. fumigatus* assimilates volatile sulfur compounds (VSCs) via cysteine synthase or homocysteine-synthase. This process is necessary for the use of VSCs as sulfur sources, since *P. aeruginosa*-derived VSCs trigger growth of *A. fumigatus* wild-type, but not of a mutant defective in cysteine synthase and homocysteine-synthase on sulfur-limiting media [79]. However, in this study a substrate was used that resembles the lung environment, indicating that this volatile-based synergy also occurs in the human lungs, although further research is still needed to confirm this. Also, the mechanistic basis of such cross-talk and its physiological relevance during co-infection remains unknown. 

Under certain conditions, *P. aeruginosa* contributes to the colonization and growth of *A. fumigatus* in CF patients. Therefore, it is possible that both pathogens work together to interfere with host immune factors, resulting in increased infection and decreased pulmonary function. Moreover, cytotoxic elastase produced by *P. aeruginosa* increases in the presence of *A. fumigatus*, leading to damage of the human lung epithelial cells, decreasing lung function and enhancing progression of the disease [80]. However, their relationship becomes more competitive with growth and the change of environmental circumstances. In a low Fe environment, *P. aeruginosa* inhibits *A. fumigatus* via several mechanisms. In an environment with high iron levels, *P. aeruginosa* enhances *A. fumigatus* growth [72]. In the presence of co-infection, *P. aeruginosa* and *A. fumigatus* do not exist in isolation; instead, they affect each other and maintain a changing synergistic relationship.

## 6. Genetic Evolution and Adaptation

In order to keep surviving in the human host, pathogenic bacteria, yeast and fungi have to adapt to the in-host environment. Adaptation to the in-host environment can occur through several mechanisms. One of these mechanisms involves epigenetic changes, where the genetic code of DNA is unchanged but the activation of certain genes is modified. Another strategy to enhance the survival chances of the pathogen is through genetic adaptation, which can be defined as the acquisition of heritable modifications. Genetic adaptation in biofilms can be a severe clinical issue, since the resistant biofilms can enhance their resistance even further through additional mutations. Genetic diversification is often more observed in biofilm populations compared to planktonic cell cultures [81]. For example, in the bacterium *Mycobacterium tuberculosis (M. tuberculosis*), the formed biofilms showed distinct characteristics compared to planktonic cultures. Whereas exposure to isoniazid and rifampicin lead to cell death in planktonic cultures, biofilms exposed to the antibiotics had a high proportion of cells (around 10%) that survived incubation with isoniazid treatment [82]. This example suggests that genetic adaptation is more likely to occur in cells of biofilms. To tackle the problem of these adaptive mutations that seem to occur at a higher rate in biofilms, studies should reveal the underlying mechanism behind genetic adaptation. This could provide new insight in the evolutionary predictability of molecular changes in certain conditions e.g., in CF patients and help to develop novel strategies against *A. fumigatus* antifungal resistance and infection. 

### 6.1. Primary and Acquired Resistance Mechanisms

Fungal pathogens can use several mechanisms to evolve resistance against antifungal drugs. These adapted resistance mechanisms can be divided into primary and acquired resistance, and an overview of these mechanisms were described previously [80]. The primary resistance mechanisms involve the structure of the formed biofilm that inhibits fungal drug susceptibility, target incompatibility through amino acid substitutions, stress response signaling, overexpression of efflux pumps that act as drug transporters, and alterations in cell wall permeability that could prevent antifungal drugs from reaching their targets [83]. 

The acquired resistance mechanisms include overexpression of the drug target and alterations in cellular pathways. Other acquired resistance mechanisms can overlap with the mechanisms of the primary resistance pathways. These include amino acid substitutions leading to reduced drug binding, signaling through stress response pathways, and upregulation of efflux pumps. Moreover, acquired resistance mechanisms can be accelerated by several factors that enhance genetic adaptation. One of the factors that can enhance genetic adaptation includes genetic plasticity, the presence of hypermutator strains, or environmental pressures, such as sudden exposure to antifungals, that can lead to strains that are more resistant to fungicides. The combination of these primary and acquired mechanisms can result in the selection of resistant microorganisms [83]. 

Cells in biofilms are able to adapt to the in-host environment and stresses in multiple ways. The altering of gene expression is a strategy where the cells are able to change the phenotypic traits in favor of survival. Another mechanism to enhance beneficial traits is through genetic adaptation, where mutations or changes in gene expression may lead to more beneficial traits that enhance the survival chances of the pathogenic fungi. Mutations may result in point mutations or can change larger parts of the genome. An overview of these genetic adaptation mechanisms will be presented in the following sections. 

### 6.2. Phenotypic Plasticity

Biofilms are able to quickly adapt to environmental changes or stresses. One of these mechanisms is the ability to rapidly change their phenotypic traits to enhance their survival chances. An example is the upregulation of efflux pumps as a result of the presence of antifungal agents. Multiple studies that were also mentioned earlier showed that efflux pumps, such as AfuMDR4, MDR1, MDR2, and MDR4, were upregulated in *A. fumigatus* biofilms when exposed to concentrations of antifungals [50,51]. Moreover, in a study where *A. fumigatus* biofilm was exposed to antimicrobial agents, it was shown that the fungus was able to adjust its transcriptome rapidly within 60 min of exposure to itraconazole [84]. This suggests that when *A. fumigatus* is challenged with a new and stressful environment, the mechanisms of phenotypic plasticity allow for the adjustment of its physiology to enable growth and reproduction. 

Phenotypic plasticity can also be triggered in cases of environmental changes in the host. Since efficient adhesion and colonization of pathogens onto surfaces are crucial steps in the process of biofilm formation, it is possible that biofilms are flexible in adapting to a variety of surfaces. A study by Shemesh et al. [85] analyzed the molecular conditions during biofilm growth of *Streptococcus mutans* (*S. mutans*) on different dental surfaces. Growth analyses showed that the bacteria formed a smoother and thicker biofilm on a hydroxyapatite surface compared to the other tested surfaces. The differentially expressed genes of *S. mutans* were analyzed through transcriptional profiling. This revealed widely based changes in the patterns of gene expression during development of the *S. mutans* biofilm on various surfaces. These results indicated that the physiological state of bacteria can be influenced by the type of substance of attachment. In another study, separate isolates of *P. aeruginosa* were exposed to the same environmental pressures, which resulted in the production of similar biofilm phenotypes. This observed independent evolution of particular biofilm phenotypic traits in non-related clinical isolates showed that parallel evolution in *P. aeruginosa* can occur in adaptation to comparable environments [86].

The same effect of parallel evolution was observed in isolates of *Burkholderia cenocepacia* (*B. cenocepacia*). In these experiments, shared selection pressure drove parallel evolution across distinct environments. The *B. cenocepacia* populations evolved for 90 days under all combinations of high or low carbon availability with selection for either planktonic or biofilm modes of growth. They found that genetic parallelism between low-carbon biofilm and low-carbon planktonic populations was very low despite shared selection for growth under low-carbon conditions, suggesting that evolution in low-carbon environments could develop stronger trade-offs between the growth of biofilm and planktonic cultures. However, they also found that a population’s fitness in a specific environment was positively correlated with the genetic similarity between that population and the populations that evolved in that specific environment. This showed that in general, evolution in similar environments led to higher levels of genetic parallelism [87]. 

### 6.3. Genomic Plasticity

Genomic plasticity allows microorganisms to adapt to environmental stresses and in-host defense mechanisms. In most cases, changes are brought to the DNA. An example of this is through horizontal gene transfer (HGT), where genetic material is transferred between microorganisms, and for example contributes to antibiotic resistance development. HGT has been detected in the genomes of *A. fumigatus*, *A. nidulans* and *A flavus* using the phyletic distribution-based bioinformatics software called HGT-Finder. These analyses support the hypothesis that secondary metabolite gene clusters evolved via HGT [88]. 

A study by Valdes et al. [89] investigated whether *A. fumigatus* was also able to acquire mutations in the case of long-term exposure to the sino-nasal environment of dogs. Through whole genome sequencing, single-nucleotide polymorphisms (SNPs) were identified in isolates from dogs suffering from sino-nasal aspergillosis. They found that in a total of 28 isolates contained on average around 45,000 non-synonymous (ns) SNPs, but six isolates from a total of three dogs contained approximately 2.5-fold more mutations. The SNP isolates formed a closely related subclade, despite the fact that they originated from 3 genotypes. This indicated that the six isolates had acquired similar genetic changes. Part of these ns-SNPs were shared, which gives the impression that they resulted from pressure in the host. The variants containing 2.5-fold more SNPs might have an increased mutation rate which could be the result of defects in DNA repair, since many mutations actually were detected in DNA-repair machinery. Together, the results of this study indicated that the mixture of variants of *A. fumigatus* found at the sino-nasal mucosal surface in the dog is a result of in-host adaptation. 

In another study where the adaptive resistance mechanisms were analyzed, thirteen *A. fumigatus* strains were isolated from a patient suffering from recurring IA over a period of two years. In all strains identical microsatellite genotypes were observed and were considered isogenic. Through whole genome comparisons they identified non-synonymous single nucleotide polymorphisms that potentially play a role in adaptation in the host. Further experiments analyzing growth, virulence, and conidiation in these strains demonstrated that *A. fumigatus* in this patient had enhanced azole resistance mechanisms and additional phenotypes that promoted fungal resistance [90]. 

Other forms of adaptation strategies of *A. fumigatus* and *C. albicans* lie in their reproduction strategies [91,92]. *A. fumigatus* is able to reproduce through asexual reproduction, sexual reproduction, and parasexual reproduction. Genomic changes can occur during the three reproduction processes. During asexual reproduction, conidiophores that are produced by hyphae can produce a great number of spores. Through every mitotic phase, there is a probability of a mutation. However, since the number of mitotic divisions is so large during asexual production, asexual spores can contribute to a large number of spontaneous mutations that can be beneficial for azole resistance mutations when exposed to azoles [93]. 

During sexual reproduction, genetic variation can occur through recombination during the meiosis process, where the genotypes of the parental strains of opposite mating types are reshuffled. In case of azole pressure, more resistant phenotypes can occur through natural selection. However, sexual production can take several months to complete, and was only demonstrated under controlled laboratory conditions. Sexual production is unlikely to occur in the human host due to required conditions [92,94]. 

The third reproduction strategy of *A. fumigatus* involves parasexual recombination. This strategy also enhances genetic diversity through the reshuffling of genes that can result in genetically different but compatible hyphae. Whereas asexual and sexual reproduction are known to occur in many microorganisms, parasexual recombination is more considered as a strategy of the fungus to generate genetic diversity than a reproduction mode. During parasexual recombination, the fungal hyphae fuse together, allowing the nuclei to fuse as well. This can result in a diploid phase in which mitotic recombination can occur, before changing back to the initial haploid stage. This form of reproduction strategy can also result in genetic variation and enabling the production of genotypes that are more adapted to the changing in-host environment [92,95]. 

### 6.4. Hypermutation

Microorganisms that undergo mutations at a high rate, also known as hypermutator strains, can be very efficient in genetically adapting to environmental stresses. These mutations can be caused by errors in DNA replication or chromosome segregation, which can result in changes in whole chromosomes or chromosome segments [96].

In a study by Mena et al. [97], it was demonstrated that *P. aeruginosa* strains in CF patients suffering from chronic respiratory infection (CRI) experience substantial genetic adaptation. In a collection of isolates, the role of hypermutation was examined in this process, since a study by Oliver et al. [98] found that one of the trademarks of CRI is the widespread occurrence of DNA mismatch repair in (MMR) system-deficient mutator strains. They found mutations in 34 genes in the study collection of 90 *P. aeruginosa* isolates, but these mutations were not homogeneous. The mutations were significantly concentrated in the mutator lineages, which represented 17% of the isolates. On an interesting note, they found that increased accumulation of mutations in mutator lineages was not a consequence of overrepresentation of mutations in genes involved in antimicrobial resistance, which was the only adaptive trait that was connected so far to hypermutation in CF patients. This demonstrated that hypermutation in DNA repair systems also plays a major role in *P. aeruginosa* genome evolution and adaptation during CRI [97]. 

Another mechanism that was found to possibly contribute to genetic adaptation in biofilms is the appearance of coding tandem repeats. Genes containing multiple coding minisatellite- and short tandem repeats are highly dynamic components of genomes. This variation in intragenic repeat number supplies the functional diversity of cell surface antigens. Through this mechanism, pathogens are able to adapt rapidly to environmental pressures and avoid the host immune system [99]. In a study by Levdansky et al. [100], the genome of *A. fumigatus* was analyzed for open reading frames (ORFs) containing tandem repeat regions, and the variability in the length of these regions in different isolates was determined. They found that expansion and contraction of coding tandem repeat regions occurred in *A. fumigatus* genes, and that a significant proportion of those genes undergoing expansion and contraction of coding repeat regions encode putative cell surface proteins. Deletion of one of these genes, Afu3g08990, which encodes the CspA protein localized at the cell wall, results in rapid germination and decreased adherence to ECM, suggesting that it plays a role in defining cell surface properties. Moreover, in a study by Fan et al. [101] disruption of the *cspA* gene resulted in an alteration of the conidial surface, a decrease in biofilm formation and less resistance to antifungal drugs. Also, internalization by lung epithelial cells was shown to be increased in the Δ*cspA* mutant, which suggests that CspA not only contributes in preserving the integrity of the cell wall, but also plays a role in biofilm establishment, response to antifungal drugs, and the uptake of *A. fumigatus*.

DNA modification through aneuploidy is another described mechanism where the fungal genome obtains a different chromosome number than the wildtype [102]. Aneuploidy is also detected in other fungal pathogens isolated from patients and the environmental setting, which suggests that variation in chromosome organization is a familiar mechanism of pathogenic fungi to quickly generate genetic diversity in the case of environmental stresses, such as exposure to antifungals [91]. Large changes in chromosomes in cases of antimicrobial pressure have also been observed in *C. albicans,* where truncation or fragmentation of chromosomes have been reported for isolates from the clinical and laboratory setting. In an isolate of *C. albicans* from a bone marrow transplant patient treated with fluconazole, truncations in the (Chr5) chromosome were observed [103]. Also, processes such as variation in copy numbers during replication and loss of heterozygosity, a cross chromosomal event where one of the alleles at the locus can get lost, may result in genomic changes and can stimulate antimicrobial resistance and virulence. However, the underlying mechanism is not fully clear [104].

## 7. Transcriptomic Studies of Biofilms

Together with gene deletion analysis and morphological studies, biofilms produced by *A. fumigatus* have been studied to a very limited extent in in vitro and in vivo model systems with microarray and RNA-sequencing (Table 2). A comparison between in vitro biofilms using microarray, RNA-sequencing and proteomic analysis was already described by Muszkieta et al. [105] and Beauvais & Latgé [106]. In these studies, gene expression was compared between *A. fumigatus* biofilms grown on solid agar medium (BF) versus submerged planktonic growth in liquid medium conditions (PL). These studies clearly indicated that aerial growth on agar medium represented a BF-like state with characteristics like reduced susceptibility for antifungals, the presence of an extracellular matrix, and changes in metabolism that were more associated with virulence. In contrast, the submerged growth system (PL) was shown to be a poor in vitro disease model. Furthermore, RNA-sequencing was shown to be a much more sensitive technique, and only around 49% of the differently expressed genes identified in microarrays were also detected in RNA-sequencing analysis. The proteomic analysis provided very limited data that could be connected to gene-expression analysis [105]. Currently, proteomic technology is much more advanced, however no new proteomic analysis of *A. fumigatus* biofilms has been described yet. 

Among the important genes that were found to be upregulated in the datasets of the studies described in Table 2 were the ones involved in production of gliotoxin (GliG, GliK) and melanin (Arb1, Arp2). Gliotoxin, for example, is an important molecule in the regulation of the host response via several mechanisms, for example the induction of apoptosis in leukocytes and the inhibition of T-cell and B-cells response [107]. Interestingly, expression of the gliotoxin and hexadehydro-astechrome clusters was also detected in mature biofilms in dogs suffering from sino-nasal aspergillosis (SNA) [89], underscoring the important role of these secondary metabolites for *A. fumigatus* growing in biofilm. 

Stress response is another crucial factor for the establishment of a successful biofilm since the fungus encounters stress in the host environment (e.g., oxidative stress, low oxygen and nutritional immunity). Mechanisms to cope with such challenges can be expected to be upregulated. For example, superoxide dismutase (SODs) are proteins that detoxify harmful reactive oxygen species (ROS), and a well-known immunoreactive SOD from *A. fumigatus* is SodA (Afu5g09240), which was upregulated in 24 h and 48 h biofilm [76], Interestingly, however, two SOD proteins, SodB (Afu4g11580), and SodM (Afu1g14550) were downregulated in 24 h and 48 h biofilms, indicating a condition dependent expression of the superoxide dismutases of *A. fumigatus*. Such a pattern is not new and was also reported previously in a study comparing the gene expression of conidia, swollen conidia, germinated conidia and mycelia [108]. Here it was shown that SodA and SodB were highly expressed in resting conidia, and SodM only on mycelium.

Transcription factors play a major role in the resistance to stress as well. It has been shown in *Aspergillus nidulans* that the transcription factor AtfA plays a critical role in the induction of many different stress-responsive genes. Its influence stretches from responses towards osmotic, heat shock- and oxidative-stress [109,110]. The importance of this gene in *A. fumigatus* biofilm is underlined by the relative high expression levels shown in all studies indicated in Table 2. In in vivo gene expression studies in mice, a role for the transcription factor PacC in development of invasive aspergillosis was described. Bronchoalveolar lavage of infected mice in a time course experiment up to 16 h post infection followed by micro-array analysis revealed insight into the early steps in fungal development which can also be regarded as initial steps of biofilm formation. PacC mutants revealed a non-invasive phenotype and were shown to remain localized at the surface of lung epithelial cells. 

**Table 2 jof-08-00048-t002:** Overview of gene expression studies of A. fumigatus biofilms. Whole genome expression studies of *A. fumigatus* biofilm growth were used to cross-reference with the PHI database [111]. No modifications of expression data or reanalysis of raw data was made in the case of Bertuzzi et al. 2014 and Gibbons et al. 2012 [112,113]. The data marked with bullet points was used to describe which genes were up- or down-regulated in that specific study. Expression thresholds for differential expression were kept as reported by the authors. Additionally, names of the categories used for Figure 2 are depicted.

	Objective	Model	Object of Analysis	Time Point/Series of Analysis after Growth	Expression Threshold for Differential Expression, and Selected Categories
Bertuzzi et al. (2014) [112]	Transcriptional profiling of Δ*pacC* ^ATCC^ mutants	In vivo	Bronchoalveolar lavage of infected mice (in vivo)	4, 8, 12 and 16 h of WT and Δ*pacC* ^ATCC^	Log2 ratios ≥ +/− 1.5 relative to ungerminated spores (only WT) Early expressed: (differentially expressed at any or all of 4, 8 and 12 h post-infection Late expressed: (differentially expressed at either or both of 12 and 16 h post-infection Mice up = Early up + late up Mice down = Early down+ late down
Bruns et al. (2010) [76]	Proteome and transcriptome analysis of PL versus BF	In vitro	Planktonic and biofilm-grown *A. fumigatus* mycelium	24 and 48 h	Biofilm 24 = Log2 ratio > ±2.3 Biofilm 48 = Log2 ratio > ±4 at 48 h
Gibbons et al. (2012) [113]	Gene expression analysis of BF versus PL	In vitro	Fungal tissue from planktonic and biofilm grown *A. fumigatus*	16 h	Log2 ratio between Biofilm and planktonic RPKM values. Biofilm 16 up = Biofilm unique (only expressed in Biofilm) 2 -fold up in Biofilm Biofilm 16 down = 2 -fold down in Biofilm

Another transcription factor known to play a role on stress response is Yap1. This transcription factor specifically regulates genes with defensive functions towards reactive oxygen intermediates (ROIs). ROIs are produced by alveolar macrophages, which have been shown to kill conidia. These alveolar macrophages are the most common immune cell that fight *A. fumigatus* in the lungs [3,114]. Yap1 was found to be down regulated in 24 h [76].

*A. fumigatus* growing in the host and biofilm will encounter nutritional immunity stress as a result of deprivation of nutrients, and zinc deprivation is one of them [112]. In the study of Bertuzzi et al. [112] it was observed that the zinc transporter ZrfB (Afu2g03860) was downregulated under neutral-alkaline conditions via PacC (Afu3g11970), suggesting that *A. fumigatus* biofilms are neutral to alkaline. Interestingly, mature biofilms in dogs suffering from sino-nasal aspergillosis (SNA) have an elevated pH of around 8 [89]. In agreement with this, it was found that the ZrfC (Afu4g09560) zinc transporter was also expressed in biofilms in SNA, which is required for zinc acquisition under alkaline conditions [115]. 

We compared the in vivo whole gene expression studies in the mice of Bertuzzi et al. [112] with the transcriptomic studies of in vitro formed biofilms [76,113] and analyzed which differentially expressed genes that are known to be involved in pathogenesis (as reported in the PHI database) were shared in these three studies (Figure 2). Remarkably few genes were found to be shared between all conditions. For example, the dataset of Gibbons et al. [113] shared upregulation of genes involved in the gliotoxin cluster (GliG and GliK) with the dataset of Bertuzzi et al. [112], and upregulation of the lectin FleA with the 48 h biofilm condition of Bruns et al. [76]. Although this analysis was made on a small set of genes, it shows that future work needs a standard biofilm assay to study biofilm formation and pathogenesis. Furthermore, the results emphasize the great plasticity of this fungus to adapt to several conditions.

## 8. Conclusions and Future Directions

Since the increased developments in medical treatments, patients suffering from severe diseases now have higher chances of survival compared to years before. However, these developments in the medical field also led to an increase of immunosuppressed patients in hospitals. These patients are more vulnerable to nosocomial infection caused by opportunistic pathogens such as *C. albicans* and *A. fumigatus*. *Candida spp.* can cause diseases such as urinary tract infections, hospital pneumonia, and biofilm formation on the surface of medical devices. *A. fumigatus* is responsible for diseases such as invasive aspergillosis, and can also infect the human host by biofilm growth on implanted medical devices. These infections are an increasing problem in the clinical setting. Infection of these pathogens are often harder to treat in the case of biofilm formation, since biofilms often possess strong antibiotic resistance. 

This review first covered the classical view of biofilm development, focusing on the different stages of biofilm formation that can be generally divided into adhesion of the conidia to host-cells, conidial germination and development of hyphae, maturation of the biofilm, and cell dispersion. Compared to *A. fumigatus*, biofilm formation in *C. albicans* is studied more broadly. Biofilm growth of *A. fumigatus* is slower compared to *C. albicans*, and studies regarding *A. fumigatus* biofilm seem to be performed more often in vitro than in vivo compared to *C. albicans* studies. Following *A. fumigatus* isolates more often in a clinical setting could help to gain more insights in the underlying mechanisms of *A. fumigatus* biofilm formation. 

The reason biofilms have such a strong antimicrobial resistance is due to the ability to rapidly adapt to their (in-host) environment. The most efficient mechanisms behind this adjustment to environmental stresses are linked to genetic adaptation. The strategy of genetic adaptation can occur on different levels. Alterations can occur in the form of phenotypic traits where certain genes can be upregulated such as efflux pumps, but in essence the fungal genome remains unchanged. 

On a deeper level of genetic adaptation, modifications can be made in the genome through point mutations, horizontal gene transfer, and errors in DNA repair systems leading to mutations that may result in beneficial traits for the survival of the fungi. Another moment in the lifecycle of the fungus that is sensitive for probable mutations is during the reproduction process. Through asexual, sexual, and parasexual reproduction, spontaneous mutations can occur that may result in helpful characteristics in antifungal resistance. At last, through processes such as hypermutation in strains, loss of heterozygosity, severe errors in DNA mismatch repair, aneuploidy and copy number variation, large, or even whole parts of the chromosome can be altered, increasing the mutation rate in these strains and therefore the chance of novel beneficial traits. 

Most research concerning *A. fumigatus* biofilm formation is focused on the underlying mechanisms in the biofilm development process. However, the problem in treating *A. fumigatus* infections lies in the ability to quickly develop antifungal resistance. Therefore, more research is needed to understand the strategies that *A. fumigatus* uses to increase its genetic diversity during infection in the environment of the human host. A strategy to investigate genetic adaptation could include performing large scale analysis with in depth sequencing methods of isolates from patients. This way, genomic modifications, large or small, can be analyzed and could give a better view of the *A. fumigatus* mutations that occur randomly or perhaps in a more controlled manner when exposed to antifungal drugs. Perhaps specific genes could be discovered that are related to genomic plasticity.

With this information, genomic changes leading to resistance could be detected easier, or even predict which specific antifungal treatments can be used best to enhance treatment efficiency. With such strategies, more targeted treatments for aspergillosis and other *A. fumigatus* related diseases could be developed and improved. In order to understand the molecular processes of biofilm formation, standard biofilm assays are required for future work which will allow more in-depth gene expression studies. It will be challenging to develop such assays that reflect in vivo biofilm growth.

## Figures and Tables

**Figure 1 jof-08-00048-f001:**
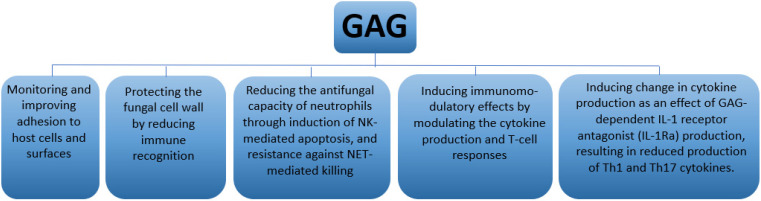
Potential roles of GAG during fungal infection.

**Figure 2 jof-08-00048-f002:**
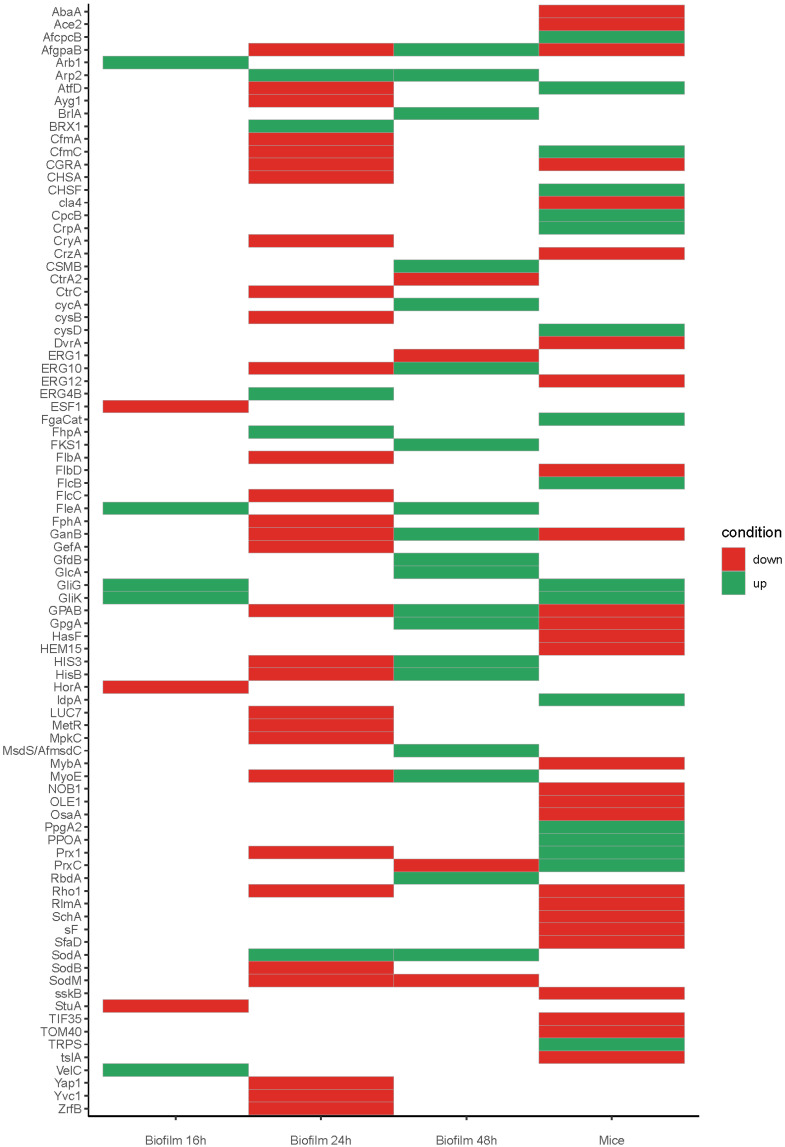
Heatmap showing the expression profiles of selected genes that are known to be involved in pathogenesis as reported by PHI database [111], expression and thresholds for differential expression was kept as reported by the authors, for mice and biofilm data, some conditions were grouped into one condition were: Mice up = Early up + late up. Mice down = Early down + late down. Biofilm 16 up = Biofilm unique (only expressed in Biofilm) 2-fold up in Biofilm. Biofilm 16 down = 2-fold down in Biofilm (Table 2). Full expression values are on Appendix A.

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
