# Peer review of "Growth of Aspergillus fumigatus in Biofilms in Comparison to Candida albicans"

_jof, 2022, doi:10.3390/jof8010048_

Round 1
Reviewer 1 Report
In the review “Growth of Aspergillus fumigatus” (jof-1526844), the manuscript is very well written and structured, almost exhaustive, and the references relevant. One point was not mentioned at all which prevents me from accepting the paper without revision: it is the fungal quorum sensing, which could be integrated in the paragraphs "Biofilm maturation", "Cell dispersion" and / or Drug resistance mechanisms through cell density.
Quorum sensing molecules are fairly well studied in Candida albicans and much less in Aspergillus fumigatus, but results have already been obtained, especially for farnesol: Dichtl K et al. Mol Microbiol 2010, 76, 1191–1204; Costa-Orlandi CB et al. JoF 2017, doi:10.3390/jof3020022; Padder SA et al. Microbiol Research 2018;210:51-8…
In addition, I have another comment, which would not have required revision, regarding persistent cells (page 7): the authors could have specified that their presence was variable within the biofilms of Candida albicans according to the isolates (Denega I et al. Antimicrob Agents Chemother. 2019 Apr 25;63(5):e01979-18. doi: 10.1128/AAC.01979-18). It is therefore not a constant resistance mechanism of the biofilm, at least for this yeast.
Typos in the manuscript:
- Incompletes reference 10, 11, 92 : journal, volume and/or pages are missing.
- Line 326: names of the authors in parentheses to be removed
Author Response
we thank the reviewer for the valuable comments which we addressed in the accompanying file

Reviewer 2 Report
General comments
- It is recommended to change the title of the paper to be: Growth of Aspergillus fumigates in biofilm in comparison to Candida albicans. As throughout the paper Aspergillus biofilm was usually compared with Candida
- Throughout the manuscript always start with talking about Aspergillus then compare with Candida and not the reverse
- The paper contain so much details about some topics not important to fungal biofilm, for example steps of GAG biosynthesis in first paragraph in page 9 and the effect of Pseudomonas aeruginosa on growth of Aspergillus in general (and not biofilm) as in page 11, please focus on Biofilm formation details
- Sometimes the authors make comparison with bacterial species such as Burkholderia cenocepacia (Lines 587-598), Vibrio cholera (lines 604-607) and Pseudomonas aeruginosa (lines 660-672). Please try to make comparisons with Candida albicans or other related fungal species such as Aspergillus nidulans or non-albicans Candida species
- The reference style are not the same in all the references throughout the reference list, please try to stick to the Journal of Fungi reference style
Specific comments
- If a paragraph followed by 2 references should be inside the same brackets, do not use separate brackets for each reference
- Titles and subtitles should be numbered
- Line 52 , of primarily: should be primarily of
- Line 326, remove the authors name after reference number [53] à not needed
- In table 1, make references in numbers and not as authors and date
- Line 706, the title transcriptome studies in biofilm should be bold
- The last title should be conclusions and future directions, and not discussion
Author Response
we thank the reviewer for the valuable comments
Reviewer 2
Comments and Suggestions for Authors
General comments
- It is recommended to change the title of the paper to be: Growth of Aspergillus fumigates in biofilm in comparison to Candida albicans. As throughout the paper Aspergillus biofilm was usually compared with Candida. Thanks, we agree and have changed the title accordingly
- Throughout the manuscript always start with talking about Aspergillus then compare with Candida and not the reverse. Thanks, We have adapted this at the start of the section 2.4 where we feel this was most obvious to do so.
- The paper contain so much details about some topics not important to fungal biofilm, for example steps of GAG biosynthesis in first paragraph in page 9 and the effect of Pseudomonas aeruginosa on growth of Aspergillus in general (and not biofilm) as in page 11, please focus on Biofilm formation details. We agree and have removed the part describing the GAG biosynthesis.
- Sometimes the authors make comparison with bacterial species such as Burkholderia cenocepacia (Lines 587-598), Vibrio cholera (lines 604-607) and Pseudomonas aeruginosa (lines 660-672). Please try to make comparisons with Candida albicans or other related fungal species such as Aspergillus nidulans or non-albicans Candida species. We have used comparisons to fungal species mostly but indeed included some very specific examples of bacterial origin since they are important as mechanistic examples eg for parallel evolution in relation to growth under low carbon conditions in either biofilms or planktonic growth (Burkholderia), adaptation via HGT (Vibrio) and the occurrence of hypermutators (Pseudomonas). We replaced the Vibrio example for Aspergillus examples of HGT. The other two examples are as far as we know not described yet for fungal species indicated by the reviewer but the mechanisms could be expected to be present in fungi.
- The reference style are not the same in all the references throughout the reference list, please try to stick to the Journal of Fungi reference style. We checked and adapted the reference list accordingly
Specific comments
- If a paragraph followed by 2 references should be inside the same brackets, do not use separate brackets for each reference. We adapted this accordingly
- Titles and subtitles should be numbered. We adapted this accordingly
- Line 52 , of primarily: should be primarily of. We adapted this accordingly
- Line 326, remove the authors name after reference number [53] à not needed. We adapted this accordingly
- In table 1, make references in numbers and not as authors and date. We adapted this accordingly
- Line 706, the title transcriptome studies in biofilm should be bold. We adapted this accordingly
- The last title should be conclusions and future directions, and not discussion. We adapted this accordingly